# Assessment of the Risk of Exotic Zika Virus Strain Transmission by *Aedes aegypti* and *Culex quinquefasciatus* from Senegal Compared to a Native Strain

**DOI:** 10.3390/tropicalmed8020130

**Published:** 2023-02-20

**Authors:** Alioune Gaye, Cheikh Fall, Oumar Faye, Myrielle Dupont-Rouzeyrol, El Hadji Ndiaye, Diawo Diallo, Paolo Marinho de Andrade Zanotto, Ibrahima Dia, Scott C. Weaver, Mawlouth Diallo

**Affiliations:** 1Pole de Zoologie Médicale, Institut Pasteur de Dakar, 36, Avenue Pasteur, Dakar BP 220, Senegal; 2Pole de Microbiologie, Institut Pasteur de Dakar, 36, Avenue Pasteur, Dakar BP 220, Senegal; 3Pole de Virologie, Institut Pasteur de Dakar, 36, Avenue Pasteur, Dakar BP 220, Senegal; 4URE Dengue et Arboviroses, Institut Pasteur de Nouvelle-Calédonie, Réseau International des Instituts Pasteur, BP 61, CEDEX, 98845 Noumea, New Caledonia; 5Microbiology Department, Institute of Biomedical Sciences, University of São Paulo (USP), São Paulo 05508020, SP, Brazil; 6World Reference Center for Emerging Viruses and Arboviruses, Institute for Human Infections and Immunity, Department of Microbiology and Immunology, University of Texas Medical Branch, Galveston, TX 77555, USA

**Keywords:** Zika virus, *A. aegypti*, *C. quinquefasciatus*, Brazil, New Caledonia, Senegal

## Abstract

Zika virus (ZIKV) shows an enigmatic epidemiological profile in Africa. Despite its frequent detection in mosquitoes, few human cases have been reported. This could be due to the low infectious potential or low virulence of African ZIKV lineages. This study sought to assess the susceptibility of *A. aegypti* and *C. quinquefasciatus* to ZIKV strains from Senegal, Brazil, and New Caledonia. Vertical transmission was also investigated. Whole bodies, legs/wings and saliva samples were tested for ZIKV by real-time PCR to estimate infection, dissemination and transmission rates as well as the infection rate in the progeny of infected female *A. aegypti*. For *A. aegypti*, the Senegalese strain showed at 15 days post-exposure (dpe) a significantly higher infection rate (52.43%) than the Brazilian (10%) and New Caledonian (0%) strains. The Brazilian and Senegalese strains were disseminated but not detected in saliva. No *A. aegypti* offspring from females infected with Senegalese and Brazilian ZIKV strains tested positive. No infection was recorded for *C. quinquefasciatus*. We observed the incompetence of Senegalese *A. aegypti* to transmit ZIKV and the *C. quinquefasciatus* were completely refractory. The effect of freezing ZIKV had no significant impact on the vector competence of *Aedes aegypti* from Senegal, and vertical transmission was not reported in this study.

## 1. Introduction

Zika virus (ZIKV), belonging to the family *Flaviviridae* and genus *Flavivirus*, is an emerging arbovirus transmitted in a zoonotic cycle between *Aedes* mosquitoes of the forest canopy and nonhuman primates in Africa. ZIKV was first isolated in Uganda from a febrile sentinel rhesus monkey (*Macaca rhesus*) in 1947 and a year later from the mosquito *A. africanus* [1].

Zika fever is known to be endemic in Africa and Asia where the virus has been detected not only in mosquitoes but also by serological evidence or sporadic human cases. The first case of human infection was described in Uganda [2]. The infection is mainly characterized by mild headache, maculopapular rash, fever, discomfort, conjunctivitis, and arthralgia [3].

Except for a few sporadic cases detected in different geographical areas, no epidemic outbreak due to ZIKV was reported before 2007, when it caused the first outbreak in Micronesia in the Yap State [4], and the outbreak was detected retrospectively in Gabon [5], followed by Cambodia in 2010, in French Polynesia and New Caledonia in 2013 [6], and the New World in 2015 [7]. With this major epidemic in 2015, cases of congenital Zika syndrome with microcephaly following infection of pregnant women, Guillain-Barre syndrome and other neurological complications associated with ZIKV infection were reported as newly recognized manifestations of the disease [8]. Imported cases were also reported in Germany, Australia, Japan, and Taiwan [9,10,11,12].

ZIKV is the most frequently isolated arbovirus from mosquitoes in West Africa. In southeastern Senegal, the virus has emerged 22 times in over 40 years of surveillance initiated since 1972 to study arbovirus biodiversity. It is also one of the few viruses that have been detected continuously in Senegalese mosquitoes for eight successive years, with more than 400 strains of ZIKV isolated from about 20 mosquito species [13].

In Africa, despite these frequent detections in common mosquitoes, few human cases have been reported in seven countries, including Uganda in 1964 (Zika locality, first case) [2]; Nigeria in 1975 (three cases in Ibadan and one in Igbo Ora) and 1971 (one case in Igbo Ora) [14]; Gabon in 2007 (one in Cocobeach and four in Libreville) [5]; Guinea Bissau in 2015 (4 in Bubaque); Cape Verde in 2015–2016 and Senegal with only serological evidence in 1965, 1967, 1970–1972, 1981, 1984, 1988, 1990, 1995, 2011 and 2015 [15].

This disparity could be explained by limitations of the human surveillance system in Africa combined with nonspecific illness typical of most human cases, which can be confused with malaria or other common infectious diseases, limited susceptibility of the African populations, or limited human pathogenicity of the virus strains circulating in Africa. However, the generally higher virulence of African strains for rodent and nonhuman primate models of human infection, compared to Asian and American strains [16], does not support this hypothesis. Moreover, several studies have shown that an African ZIKV strain is more infectious for *A. aegypti* than Asian or American strains using the same viral titer [17,18].

Another hypothesis is that the low vectorial capacity of the African mosquito populations, especially those that frequently bite humans, limits human infections. Although outside of Africa, *A. aegypti* is considered the major epidemic vector of ZIKV, a previous study evaluating the vector competence of different mosquitoes (*A. aegypti, A. luteocephalus*, *A. unilineatus* and *A. vittatus*) from Senegal showed that *A. aegypti* was incompetent to transmit ZIKV despite its high susceptibility to infection [19]. This hypothesis is supported by phylogenetic studies that described the existence of three lineages of ZIKV, namely West African, East African and Asian [20,21,22], revealing that only the Asian strains are associated with severe disease. ZIKV strains circulating in the Americas are of Asian origin. Further, the unique outbreak reported in Africa in Cape Verde in 2015–2016 was attributed to imported Asian ZIKV strains from Brazil [23] and led to 7580 suspected Zika infection cases and 18 microcephaly cases.

Therefore, a major concern for Africa is whether *A. aegypti* could competently establish an epidemic transmission if any of these exotic and emerging ZIKV variants were introduced. In addition, *C. quinquefasciatus*, whose vectorial capacity for ZIKV is still debated due to conflicting results, would be a good candidate for transmission in urban areas due to its abundance and anthropophilic behavior in some locations.

Recent studies have provided evidence of natural ZIKV infection of *C. quinquefasciatus* in Recife, Brazil, as well as the ability of *C. quinquefasciatus* from Recife and China to experimentally transmit ZIKV [24,25]. However, several other studies conducted have shown the incompetence of this species [26,27,28,29].

In order to address these questions, experimental infections were performed using *A. aegypti* and *C. quinquefasciatus* from Senegal with local (ZIKVArD132912) and exotic ZIKV strains from Brazil (ZIKVT3FBra) and New Caledonia (ZIKVNC) representing the West African, Brazilian and Asian lineages, respectively. We also assessed the vertical transmission of ZIKV by *A. aegypti* females since this transmission mode has been demonstrated for ZIKV and many other arboviruses, both in the laboratory and in the field [30,31,32,33].

## 2. Materials and Methods

Mosquito species. In this study, we tested a population of *A. aegypti* from Dakar (14°43′29″ N −17°28′24″ W) and a population of *C. quinquefasciatus* from Barkedji (15°16′50.242″ N −14°51′54.751″ W). Larvae and pupae were collected from the field. Adults were reared in the laboratory at 27 ± 1 °C and a relative humidity of 70–75%, with a 12 h photoperiod. Females (F0) were fed several times with guinea pigs’ blood to obtain F1 generation eggs. After hatching, larvae were reared at 29 ± 1 °C to obtain F1 adults, which were used in this study. F1 females were fed only with a 10% sucrose solution to avoid potential issues due to antibodies in whole blood.

ZIKV stock preparation. The ZIKV strains used in this study were from the Institut Pasteur Dakar (IPD) biobank, including T3F and NC isolated from humans in Brazil and New Caledonia, respectively, and ArD132912 from a mosquito pool in Senegal. The Brazilian lineage strain (ZIKVT3FBra) was passaged once on C6/36 cells, the New Caledonian strain (ZIKVNC), representing the Asian lineage, twice on Vero cells, and the African lineage strain (ZIKVArD132912) twice on C6/36 cells.

The virus stocks used to infect mosquitoes were prepared by inoculation onto C6/36 cells for ZIKVT3FBra and ZIKVArD132912 and onto Vero cells for ZIKVNC. Titration was also performed on C6/36 and Vero cells. The cells were maintained in Leitbovitz 15 (L-15) culture medium supplemented with 10 and 5% FBS, respectively, for C6/36 and Vero cells.

Mosquito oral infections procedure. Three- to five-day-old female F1 generation mosquitoes were placed into cardboard containers and sucrose-starved for 48 h before being allowed to take an infectious blood meal with the artificial feeding system described by Rutledge (1964) using mouse skins as membranes. The infectious blood meal contained a 33% volume of washed rabbit erythrocytes and a 33% volume of viral suspension supplemented with a 20.9% volume of fetal bovine serum (FBS), a 2.5% volume of adenosine triphosphate (ATP) at a final concentration of 0.005 M as a phagostimulant, and a 10% volume of sucrose at a final concentration of 0.03 M.

In the second series of experiments, we adopted the same approach described by Weger-Lucarelli J et al. [34], who demonstrated that infection, dissemination and transmission were higher with a ZIKV strain freshly harvested from Vero cells compared to rates obtained with virus stocks stored at −80 °C. *Aedes aegypti* and *C. quinquefasciatus* mosquitoes were fed with infectious blood meals containing ZIKVT3FBra or ZIKVArD132912 that were harvested directly from incubated C6/36 cells or frozen for one week at −80 °C.

Mosquitoes were exposed to the ZIKV at different concentrations, as shown in Table 1, Table 2 and Table 3. The time of exposure for the blood meal was limited to 1 h. Then mosquitoes were cold-anesthetized. Only fully engorged specimens were selected and transferred to cardboard containers. They were then fed with 10% sucrose and incubated at 27° ± 1 °C, with a relative humidity of 70–75% and a photoperiodicity of 12:12 for extrinsic incubation. Two replicates were done using frozen ZIKV, and one replicate using ZIKV freshly harvested from cells.

A set of mosquitoes was randomly collected at 5-, 10-, 15-, 20-, and 25-days post-exposure (dpe), cold-anesthetized and dissected. Their legs and wings were removed and transferred individually into separate tubes, and the proboscis was inserted into a capillary tube containing 1–2 μL of FBS for salivation for up to 30 min. This method has been proven for saliva production and guaranteed that viral particles remain infectious in the collected saliva of competent mosquitoes [35]. After salivation, each mosquito body (whole body except legs and wings removed) and saliva sample was put into a separate tube and stored separately at −80 °C for detection and quantification of ZIKV by real-time RT-PCR.

To study the vertical transmission of both Brazilian and Senegalese ZIKV strains, 100 *A. aegypti* females exposed to infectious blood meals were selected and separated into 5 cages of 20 individuals each. In addition, 22 females exposed to the Senegalese strain were also separated individually into small cardboard containers to follow the offspring of each female mosquito. In each cage and cardboard container, a Petri dish containing wet cotton was introduced for egg collection. After completing the first gonotrophic cycle, females were allowed to take non-infectious blood for clutches of the 2nd and 3rd cycles. At the end of the third cycle, i.e., 27 days after exposure to an infectious blood meal (dpe), mosquitoes were sampled and tested individually by real-time PCR. Only the offspring of positive mosquitoes reared individually were selected for assay, but all eggs obtained were considered when determining infection rates. Eggs from each of the 20 positive individuals or batches of mosquito pools were hatched separately. The offspring of the corresponding adult were sampled over time at days 1, 5, 10 and 15 days post-emergence (dpem), pooled at up to 10 individuals and tested by real-time PCR.

Virus detection in mosquitoes. All mosquito bodies, as well as the wings/legs from infected bodies and saliva of mosquitoes with infected wings/legs, were homogenized in 400 μL of L-15 medium containing 5% of FBS before centrifugation for 2 min at 12,000 rpm at 4 °C to separate virus supernatant and debris. The viral RNA was extracted from 140 µL of supernatant using the QIAamp Viral RNA Extraction Kit (QIAgen, Heiden, Germany), according to the manufacturer’s protocol. Amplification was performed by real-time quantitative polymerase chain reaction (RT-qPCR) using the QuantiTect Probe RT-PCR Kit (Qiagen SABiosciences Corporation, Enzymatics Inc. Cat No./ID: 204443, Hilden, Germany) and a set of primers and probes described by Faye et al. [36] using an ABI7500 Fast instrument (Applied BioSystems, Foster City, CA, USA). Briefly, the amplification conditions were as follows: reverse transcription of the viral RNA −10 mn at 50 °C, denaturation and enzyme activation −15 mn at 95 °C, followed by cycling step (45 cycles) of −15 s at 95 °C and 1 mn at 60 °C. The results were interpreted as per kit instructions and analyzed.

Data analysis. Detection of ZIKV in the mosquito body without infection of the wings/legs was considered a non-disseminated infection (infection limited to the midgut), whereas the presence of the virus in both the mosquito body and wings/legs indicated a disseminated infection. The potential transmission rates, estimated by the number of mosquitoes with positive saliva among the total number of disseminated infections, were calculated for each species and each dpe. The rates obtained were compared using Fisher’s exact test. Statistical tests were performed using R v. 2.15.1 (R Foundation for Statistical Computing, Vienna, Austria) [37]. Differences were considered statistically significant at *p* < 0.05.

## 3. Results

### 3.1. The Zika Senegalese Strain versus Exotic Zika Virus Strains

All ZIKV strains used in this first study were frozen at −80 °C. Experiments with *A. aegypti* at 15 dpe showed that ZIKVArD132912 produced a significantly higher infection rate than the exotic strains, with 52% versus 10% and 0% when compared to the Brazilian ZIKVT3FBra (*p* = 0.0000001) and New Caledonian ZIKVNC strains (*p* = 0.00001), respectively (Figure 1).

Comparison between ZIKVT3FBra and ZIKVNC for *A. aegypti* showed important variations in infection rates. Indeed, 16% of infection was obtained for ZIKVT3FBra at 5 dpe, which decreased to 3.33% at 10 dpe (*p* = 0.04). However, ZIKVNC produced statistically similar infection rates of 10% and 20% at 5 dpe and 10 dpe, respectively (*p* = 0.37). The differences between exotic strains were not statistically significant (*p* = 0.35 at 5 dpe and *p* = 0.054 at 10 dpe). At 15 dpe, no mosquito tested following exposure to the ZIKVNC strain was infected.

Only the Brazilian (ZIKVT3FBra) and African (ZIKVArD132912) strains produced disseminated infection of *A. aegypti,* and the rates were relatively high at 15 dpe with 33% and 42%, respectively. At 5 dpe, the ZIKVT3FBra strain produced dissemination infections of *A. aegypti* at a rate of 25%. No transmission has been recorded regardless of the ZIKV strain tested.

In contrast to *A. aegypti*, *C. quinquefasciatus* mosquitoes, as shown in Table 1, were completely refractory to infection by all ZIKV strains tested. All 699 *C. quinquefasciatus* tested negative for ZIKV infection.

### 3.2. Infection by Freshly Harvested versus Frozen ZIKV Stocks

The experiments with freshly harvested ZIKV strains from cell cultures (Figure 2) showed lower infection rates at 20 dpe with ZIKVT3FBra (*p* = 0.02) and at 25 dpe with ZIKVARD132912 strain (*p* = 0.03) than those obtained with stocks frozen for one week (Figure 3). While dissemination rates were higher with freshly harvested ZIKVT3FBra strain and frozen ZIKVARD132912 strain, these differences were not statistically significant (*p* > 0.05).

No saliva samples were positive regardless of the strain used or freshly harvested versus frozen, suggesting virus status does not impact the infectivity of *A. aegypti*.

The results of the *Cx quinquefasciatus* mosquito experiments are summarized in Table 2 for ZIKV strains harvested directly from incubated C6/36 cells and in Table 3 for ZIKV strains frozen for one week at −80 °C.

### 3.3. Vertical Transmission

Table 4 shows the infection rates of female *A. aegypti* mosquitoes exposed to the Brazilian and Senegalese strains of ZIKV and used for the vertical transmission study. Infection rates of offspring obtained from these infected females are shown in Table 5.

We did not detect the presence of ZIKV in the offspring derived from any of the five batches of female mosquitoes for any of the virus strains, even with the Senegalese strain, for which the infection rates of the female parents reached 92%. Similarly, all offspring from the 20 separately reared infected female mosquitoes tested negative for ZIKV. Based on these data, we concluded that there is likely little or no vertical transmission of ZIKV in Senegalese *A. aegypti*.

## 4. Discussion

We highlighted through this study crucial information on the susceptibility of Senegalese mosquito vectors to infection with ZIKV. Even some minor limitations could be pointed out, (i) such as virus stocks produced on different cell lines, (ii) using different virus titers for different ZIKV strains.

The results obtained from our first experiments demonstrated that, *A. aegypti* from Dakar were susceptible to infection by both native and exotic strains of ZIKV. However, they were more susceptible to the strain isolated from Senegal. Similar results were observed with *A. aegypti* from Cape Verde islands, which were significantly more susceptible to the autochthonous DENV-3 isolated from a patient in 2009 than to the reference DENV-2 strain isolated in Thailand [38]. Although there is a possible viral lineage effect, this may suggest a specific interaction between virus genotype and vector genotype previously described for DENVs [39]. In addition, the infection rate with the Brazilian ZIKV decreased significantly between 5 and 10 dpe (*p* = 0.04) and the New Caledonian ZIKV, from 10 to 15 dpe (*p* = 0.03). This drop in infection rates from 5 to 10 dpe or 10 to 15 dpe has been observed previously with *A. aegypti* populations from Senegal [19] and *A. aegypti* [40] and *A. albopictus* [41] populations from Singapore. Moreover, the same *A. aegypti* population from Senegal showed the same profile with a decrease in infection rates for DENV-1 and DENV-3 [42]. This decrease could be due to the response of the mosquito’s immune following the invasion of the virus [43]. Mosquitoes, as the vectors of several diseases, are susceptible to pathogen infection during their life cycles and use the innate immune system to fight against it. The innate immune system, including the production of antimicrobial peptides and lysozymes, phagocytosis, and melanization, plays a significant role in limiting viruses to a non-lethal level [44]. A more recent study showed that ZIKV-induced RNA interference response in *A. aegypti* [45]. This suspicion that the immune system controls the infection is strongly supported by the dissemination rate of 25% observed at 5 dpe, followed by the complete absence of dissemination observed at 10 dpe (when the decline in infection rate was noted for ZIKV T3FBra strain). However, for the New Caledonian strain no dissemination was reported despite high viral titer, which may be due to a lack of adaptation between the virus strain and the mosquito population circulating in different areas.

A previous study showed that long-term (>one week) freezing of ZIKV reduced infection rates in *A. aegypti* mosquitoes [34]. However, in this study, for ZIKV T3FBra and ZIKV ARD132912 strains, infection rates were higher after one week of freezing. For dissemination rates, no statistical difference was observed between fresh and frozen strains. Freezing for one week may be too short to induce the loss of infectivity, similar to previous reports for short-term freezing (4 h), which did not impact infectivity compared to more extended storage (>one week) at −80 °C [34]. Generally, the effect of freezing ZIKV on infectivity for *A. aegypti* from Senegal was not significant. Further studies, including different freezing times from one week to several months, are required to clarify this aspect.

The infection and dissemination rates obtained with the ZIKVARD132912 strain between the two experiments appeared to demonstrate a dose dependency (Figure 1, Figure 2 and Figure 3 and Table 1, Table 2 and Table 3). However, the low titers of 10^4^ and 10^5^ PFU/mL do not explain the low infection and dissemination rates or the lack of transmission we obtained. The same titer of 10^4^ FFU/mL for the Senegalese ZIKV strain has previously shown infection, dissemination and transmission rates of up to 80%, 70% and 40% for *A. aegypti* mosquitoes from Brazil and 40%, 30% and 10% for *A. aegypti* mosquitoes from Texas, USA. This viral titer also showed infection, dissemination and transmission rates of 100%, 100% and 20%, respectively, for *A. aegypti* mosquitoes from the Dominican Republic. Additionally, a blood meal titer of 10^5^ FFU/mL for mosquitoes both from Brazil and the Dominican Republic showed transmission rates of nearly 80% [46]. Moreover, the peak of ZIKV viremia in humans was estimated at a mean concentration of 7.3 × 10^4^ FFU/mL [47]. These data suggest that the low infection and dissemination rates we observed are due to vector incompetence rather than virus titer.

Furthermore, African *A. aegypti* from Gabon, Cameroon and Uganda have been shown to be less susceptible than exotic *A. aegypti* from Thailand, Cambodia, Colombia, Guadeloupe, and Guiana to infection for ZIKV strains from Cambodia, Polynesia, Philippines, Puerto Rico, Thailand and Senegal [17]. Likewise, several vector competence studies have shown that *A. aegypti* from West Africa are more refractory for arboviruses like dengue and yellow fever viruses compared to *A. aegypti* from America or Asia [48,49,50].

Very few studies have demonstrated the ability of *C. quinquefasciatus* populations (only from Brazil [24] and China [25]) to experimentally transmit ZIKV. *C. quinquefasciatus* is an attractive secondary vector candidate for urban transmission because of its abundance and anthropophilic nature in some locations. However, the vast majority of published studies fail to provide evidence for the significant role of this species in ZIKV transmission.

However, given the potential of this vector, the most abundant mosquito species in many tropical urban locations, we tested the *C. quinquefasciatus* population of Senegal to assess the risk of introducing an exotic strain. In contrast to the results obtained with the populations from Brazil and China [24,25], all the mosquitoes we tested were completely refractory (Table 1, Table 2 and Table 3). Our results are similar to many other studies where *C. quinquefasciatus* was found to be incompetent for all ZIKV strains tested [26,27,28,29]. The lack of susceptibility could be due to the adaptation of the virus strain and the mosquito genotype.

In addition to epidemic transmission, there is a concern about vertical transmission in *A. aegypti* in Africa. Our results showed that infection of both ZIKV strains in *A. aegypti* from Dakar (Table 4) was not vertically transmitted to the next generation (Table 5). These results differ from those obtained in a recent study in which a population of *A. aegypti* from Thailand was able to transmit ZIKV to the next generation but at a very low rate of 0.3% (1/290) [30]. Given this very low rate, the absence of vertical transmission in our study with the Senegalese *A. aegypti* population may be due to the small number of specimens tested (*n* = 283). Moreover, *A. aegypti* populations from other biogeographic areas also showed very low rates of vertical transmission of flaviviruses, with rates ranging from 0.21% (1: 472) to 0.15% (1: 632) for yellow fever virus (VFJ) [31], from 0.24% (1/401) to less than 0.0005% (1/1700) for DENVs [32], and from 1.61% (1/62) to 1.38% (1/72) for WNV [33]. Both the complete absence of ZIKV transmission indicated by the lack of infected mosquito saliva and the lack of vertical transmission may be indicators of the incompetence of these vectors to transmit ZIKV. Therefore, further investigations need to be conducted about genetic factors (existence or not of barriers for the virus) or mosquito-virus-microbiome interactions that may be the cause of the incompetency of Senegalese vectors to transmit ZIKV.

## Figures and Tables

**Figure 1 tropicalmed-08-00130-f001:**
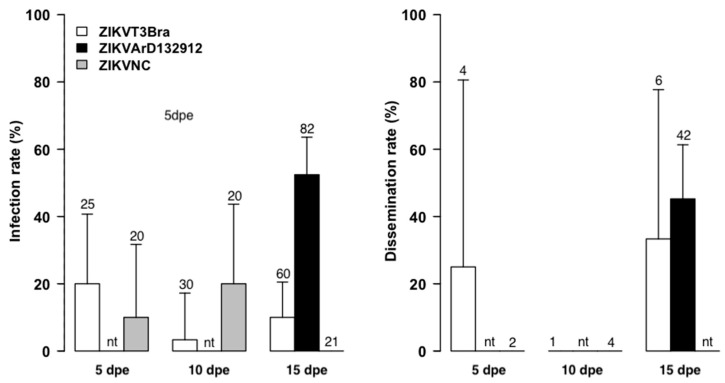
Infection and Dissemination Rates at 5, 10, and 15 dpe for *A. aegypti* from Senegal orally exposed to 10^4^, 10^6^ and 10^6^ PFU/mL of ZIKV strains isolated from Brazil (ZIKVT3FBra), New Caledonia (ZIKVNC) and Senegal (ZIKV132912), respectively. Values above each column mean the total number tested, nt = not tested.

**Figure 2 tropicalmed-08-00130-f002:**
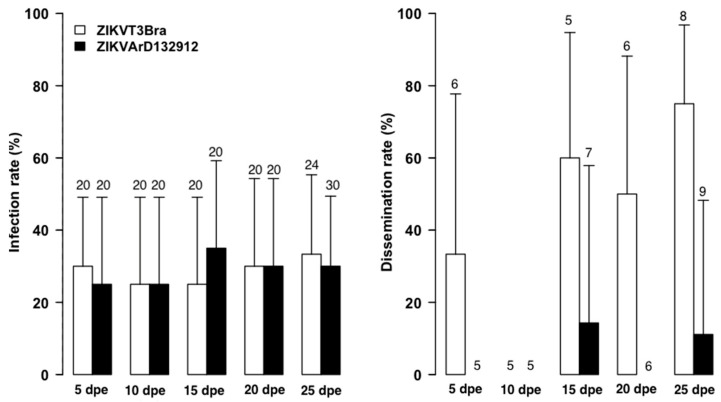
Infection and Dissemination Rates at 5, 10, 15, 20 and 25 dpe for *A. aegypti* from Senegal orally exposed to 10^4^ PFU/mL of Brazilian ZIKV (ZIKVT3FBra) and Senegalese strains (ZIKV132912) freshly harvested from C6/36 cells. Values above each column mean the total number tested, nt = not tested.

**Figure 3 tropicalmed-08-00130-f003:**
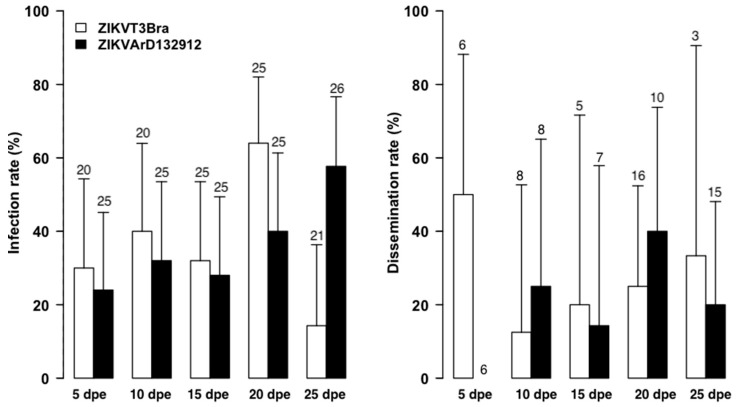
Infection and Dissemination Rates at 5, 10, 15, 20 and 25 dpe for *A. aegypti* from Senegal orally exposed to 10^5^ PFU/mL for Brazilian ZIKV (ZIKVT3FBra) and 10^4^ PFU/mL for Senegalese strains (ZIKV132912) frozen for a week at −80 °C. Values above each column mean the total number tested, nt = not tested.

**Table 1 tropicalmed-08-00130-t001:** Infection rates of *C. quinquefasciatus* orally exposed to ZIKV strains.

Zika Strain	Blood-Meal Titers (PFU/mL)	Infection Rates		
		5 dpe	10 dpe	15 dpe
ZIKVT3FBra	2.5 × 10^4^	0/30	0/30	0/37
ZIKVArD132912	5 × 10^6^	0/20	0/20	0/28
ZIKVNC	3.75 × 10^6^	0/30	0/30	0/43

dpe: day post-exposure.

**Table 2 tropicalmed-08-00130-t002:** Infection rates of *C. quinquefasciatus* orally exposed to ZIKV strains freshly harvested from cells.

Zika Strains Freshly Harvested from Cells	Blood-Meal Titers (PFU/mL)	Infection Rates		
		15 dpe	20 dpe	25 dpe
ZIKVT3FBra	7.5 × 10^4^	0/35	0/35	0/40
ZIKVArD132912	3 × 10^4^	0/30	0/30	0/25

**Table 3 tropicalmed-08-00130-t003:** Infection rates of *C. quinquefasciatus* orally exposed to ZIKV strains frozen for one week.

Zika Strains Frozen a Week	Blood-Meal Titers (PFU/mL)	Infections Rates			
		10 dpe	15 dpe	20 dpe	25 dpe
ZIKVT3FBra	1.35 × 10^5^	0/30	0/30	0/30	0/35
ZIKVArD132912	3 × 10^4^	0/25	0/25	0/25	0/36

**Table 4 tropicalmed-08-00130-t004:** Infection rates of females *A. aegypti* exposed to an infectious blood meal for the study of vertical transmission of ZIKV.

N° Batch	Infection Rates (%)	
	ZikV T3 F BRA	ZikV ArD 132912
Batch 1	3/18 (16.6)	12/13 (92.3)
Batch 2	0/15 (00)	9/15 (60)
Batch 3	4/14 (28.5)	11/13 (84.6)
Batch 4	2/6 (33.3)	7/9 (77.7)
Batch 5	3/10 (30)	11/13 (84.6)
Separated females	NA	20/22 (90.9)
Total	12/63 (19.04%)	70/85 (82.35%)

**Table 5 tropicalmed-08-00130-t005:** Infection rates of *A. aegypti* offspring from females infected with the Senegalese and Brazilian strains of ZIKV at different days after emergence.

Days Post-Emergence	Infection Rates of Offspring from Infected Females (%)		
	Eggs from Females in Batches		Eggs from Separated Females
	ZIKV T3FBra	ZIKV ARD132912	ZIKV ARD132912
0	0/20 (00)	0/20 (00)	0/10 (00)
5	0/20 (00)	0/20 (00)	0/15 (00)
10	0/40 (00)	0/40 (00)	0/15 (00)
15	0/30 (00)	0/35 (00)	0/18 (00)

## Data Availability

All the data is available in the manuscript.

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
