# Peer review of "Assessment of the Risk of Exotic Zika Virus Strain Transmission by Aedes aegypti and Culex quinquefasciatus from Senegal Compared to a Native Strain"

_tropicalmed, 2023, doi:10.3390/tropicalmed8020130_

Round 1
Reviewer 1 Report
In this work, Gaye and authors have assessed the risk of exotic ZIKV strain by two different species of mosquitoes. I have 2 major concerns:
1. the data presented in figure 1 is generated using 3 strains of ZIKV, my concern is the virus titer used as a meal for feeding mosquitoes. It is a big difference among different strains. How does this affect the infection and dissemination rate. Did author performed any study using same viral titre or there is any specific reason for these differences.
2. in figure 2 and 3 also, different titres are used for fresh and frozen samples ZIKVT3Bra strain. these two datasets can not be compared with a log difference.
3. The data in figure 1 with figure 2 and 3 does not correlate. in figure 1, the viral titre used for ZIKV132912 strain is 5X106 and in others its 104, still in Fig 1. infection and transmission was observed at 15 dpe while with low titres it is observed strating at 5 Dpe.
Author Response
Summary of corrections and changes-in the manuscript “Assessment of the risk of exotic Zika virus strain transmission by Aedes aegypti and Culex quinquefasciatus from Senegal compared to a native strain” (tropicalmed-2182029)
NOTE: We thank the reviewer for his constructive comments and suggestions. Our responses to each issue raised are outlined below in bold. Corrections were made in the paper and we have given some answers to reviewer’s inquiries accordingly. The changes made in the revision have been indicated (lines or figure).
Reviewer's Responses to Questions:
Reviewer 1
Comments and Suggestions for Authors
In this work, Gaye and authors have assessed the risk of exotic ZIKV strain by two different species of mosquitoes. I have 2 major concerns:
- the data presented in figure 1 is generated using 3 strains of ZIKV, my concern is the virus titer used as a meal for feeding mosquitoes. It is a big difference among different strains. How does this affect the infection and dissemination rate. Did author performed any study using same viral titre or there is any specific reason for these differences.
Response: Thanks to the reviewer for this pertinent concern. We have discussed difference among different strains and consider that infection and dissemination rates seemed to be titer dependent. But transmission was not observed even for high virus titer. We integrate in the main text the following sentence in lines 343-345 for more clarity “The infection and dissemination rates obtained with the ZIKVARD132912 strain between the two experiments appeared to demonstrate a dose dependency (Figures 1-3 and Tables 1-3).” We also discussed, in lines 345-356, that the absence of transmission cannot be correlated to the titers: “However, the titers of 104 and 105 PFU/ml do not explain the low infection and dissemination rates or the lack of transmission we obtained, since the peak of ZIKV viremia in humans were estimated at mean concentration of 7.3x104 FFU/ml [1]. In addition, the same titer of 104 FFU/ml for the Senegalese ZIKV strain has previously shown infection, dissemination and transmission rates of up to 80%, 70% and 40% for Ae. aegypti mosquitoes from Brazil and 40%, 30% and 10% for Ae. aegypti mosquitoes from Texas, USA. This viral titer showed also infection, dissemination and transmission rates of 100%, 100% and 20%, respectively, for Ae. aegypti mosquitoes from Dominican Republic. Additionally, a blood meal titer of 105 FFU/ml for mosquitoes both from Brazil and Dominican Republic showed transmission rates of nearly 80% [2]. These data suggest that the low infection and dissemination rates we observed are due to the vector incompetence rather than virus titer.”
- in figure 2 and 3 also, different titres are used for fresh and frozen samples ZIKVT3Bra strain. these two datasets can not be compared with a log difference.
Response: We understand the reviewer concern about the log difference between titers used for fresh and frozen ZIKVT3Bra strain, but here the main aim was to see if the absence of transmission in the previous study is due to the use of frozen ZIKV strain given the result obtained by Weger-Lucarelli et al., 2016 where fresh ZIKV strains were more infectious for Ae. aegypti. So, in our study even fresh ZIKV strain produced the same result in Ae. aegypti: a disseminated infection but no transmission.
- The data in figure 1 with figure 2 and 3 does not correlate. in figure 1, the viral titre used for ZIKV132912 strain is 5X106and in others its 104, still in Fig 1. infection and transmission was observed at 15 dpe while with low titres it is observed strating at 5 Dpe.
Response: We strongly consider the reviewer remarks, the reason why, we clearly express this aspect as a potential limit of this study in the first paragraph of discussion “We highlighted through this study, crucial information on susceptibility of Senegalese mosquito vectors to infection with ZIKV even some minor limitations could be pointed out like: i) the virus stocks produced on different cell lines, ii) using different virus titers for different ZIKV strains” see lines 291-294. Concerning the correlation, the different figures were not made simultaneously. The figure 1 was made firstly to test if Ae. aegypti can transmit exotic ZIKV strains with a native strain, and figures 2 and 3 to test if using ZIKV freshly harvested from cells can improve the vector competence and induce transmission by saliva. Furthermore, we need to clarify that transmission was not observed. Infection and dissemination started at 5 dpe with the low titers of ZIKV from Brazil, and regarding the high titers of ZIKVs, the strain from New Caledonia did not induce dissemination and that from Senegal was tested only at 15 dpe.
We thank the reviewer for the attention he gives to our manuscript. We have taken into account of his preoccupation and have addressed his different concerns in the revised version; his remarks have improved the quality of the paper.
References:
- Lequime, S.; Dehecq, J.-S.; Matheus, S.; de Laval, F.; Almeras, L.; Briolant, S.; Fontaine, A. Modeling Intra-Mosquito Dynamics of Zika Virus and Its Dose-Dependence Confirms the Low Epidemic Potential of Aedes Albopictus. PLoS Pathog. 2020, 16, e1009068, doi:10.1371/journal.ppat.1009068.
- Roundy, C.M.; Azar, S.R.; Rossi, S.L.; Huang, J.H.; Leal, G.; Yun, R.; Fernandez-Salas, I.; Vitek, C.J.; Paploski, I.A.D.; Kitron, U.; et al. Variation in Aedes Aegypti Mosquito Competence for Zika Virus Transmission. Emerg. Infect. Dis. 2017, 23, 625–632, doi:10.3201/eid2304.161484.

Reviewer 2 Report
The susceptibility of mosquitoes Aedes aegypti and Culex quinquefasciatus to Zika viruses strains from Senegal, Brazil, and New Caledonia were investigated. Results indicates of the incompetence of those mosquitoes to transmit ZIKA. The conclusion is based on the fact that no saliva samples and the offspring derived from infected female mosquitoes were positive. However, the question remains - is there a guarantee that the sensitivity of the method was sufficient to detect the virus in these cases?
Technical note
In Figures 1, 2, and 3, columns with value 0 must be labeled somehow on the x-axis, otherwise the result is difficult to interpret.
Author Response
Summary of corrections and changes-in the manuscript “Assessment of the risk of exotic Zika virus strain transmission by Aedes aegypti and Culex quinquefasciatus from Senegal compared to a native strain” (tropicalmed-2182029)
NOTE: We thank the reviewer for his constructive comments and suggestions. Our responses to each issue raised are outlined below in bold. Corrections were made in the paper and we have given some answers to reviewer’s inquiries accordingly. The changes made in the revision have been indicated (lines or figure).
Reviewer's Responses to Questions:
Reviewer 2
Comments and Suggestions for Authors
The susceptibility of mosquitoes Aedes aegypti and Culex quinquefasciatus to Zika viruses strains from Senegal, Brazil, and New Caledonia were investigated. Results indicates of the incompetence of those mosquitoes to transmit ZIKA. The conclusion is based on the fact that no saliva samples and the offspring derived from infected female mosquitoes were positive. However, the question remains - is there a guarantee that the sensitivity of the method was sufficient to detect the virus in these cases?
Response: As suggested the reviewer, we provided proof of the sensitivity of the method for virus detection. We added the following sentence to discuss the efficiency of the saliva collection method: “This method has been proven for saliva production and guaranteed that viral particles remain infectious in collected saliva of competent mosquitoes”. See lines, 147-149.
Concerning the result of offspring tests, we used the same detection method (real time PCR) than the infection, dissemination and transmission for all experiments. This suggests that the absence of positive mosquito in the offspring is due to the absence of vertical transmission.
Technical note
In Figures 1, 2, and 3, columns with value 0 must be labeled somehow on the x-axis, otherwise the result is difficult to interpret.
Response: We thank the reviewer for this remark. Figures 1, 2, and 3 have been remade and we removed the transmission sections given all these rates are 0%. We added the following sentence to summarize transmission results (lines 213-214), “No transmission has been recorded regardless ZIKV strain tested.”. All columns have been labeled with the number tested even columns with value 0, according to the reviewer suggestion. Changes can be found in lines 197-199, 232-234, 238-240.
We thank the reviewer for the attention he gives to our manuscript. We have taken into account of his preoccupation and have addressed his different concerns in the revised version; his remarks have improved the quality of the paper.

Reviewer 3 Report
1. Figure 1: How many replicates were there in the experiment? Include the information in the Methods section.
2. Figure 3: Rephrase the caption. The use of "respectively" does not quite fit the content. "Infection .... 25 dpe. Ae. aegypti was exposed orally to ...., respectively".
3. Line 267: "such as ... different glycosylation. Any evidence or proof?
4. Line 268: "virus stock lower than at least some human viremia level" Any evidence or proof of the comparison?
5. Line 269: "virus titers different..." How is this information crucial?
6. Lines 284-285: "suspicion..." Suggest to add more discussion on the immune mechanism of Ae. mosquito that is involved in controlling the virus infection.
7. Lines 287-291: Results were reiterated instead of critical discussion on the observation/data. Improve these lines. Authors may explain why the decline occurred alongside similar evidence from previous studies.
8. Lines 306-310: Split the information into separate sentences so that they clearly present the ideas.
9. Lines 316-319: Any specific reasons why exotic Ae. mosquitoes are more competent?
10. Lines 333: Future studies in what perspectives? Compatibility of receptors of the mosquitoes with the virus?
11. Grammatical errors: as attached, lines 337, 339, and 347.

Author Response
Summary of corrections and changes-in the manuscript “Assessment of the risk of exotic Zika virus strain transmission by Aedes aegypti and Culex quinquefasciatus from Senegal compared to a native strain” (tropicalmed-2182029)
NOTE: We thank the reviewer for his constructive comments and suggestions. Our responses to each issue raised are outlined below in bold. Corrections were made in the paper and we have given some answers to reviewer’s inquiries accordingly. The changes made in the revision have been indicated (lines or figure).
Reviewer's Responses to Questions:
Reviewer 3
Comments and Suggestions for Authors
- Figure 1: How many replicates were there in the experiment? Include the information in the Methods section.
Response: As recommended by the reviewer, the following information has been included in lines 142-143 “Two replicates were done using frozen ZIKV and one replicate using ZIKV freshly harvested from cells”.
- Figure 3: Rephrase the caption. The use of "respectively" does not quite fit the content. "Infection .... 25 dpe. Ae. aegypti was exposed orally to ...., respectively".
Response: As recommended by the reviewer the Figure 3 caption was rephrased (lines 240-242), “Figure 3. Infection and Dissemination Rates at 5, 10, 15, 20 and 25 dpe for Ae. aegypti from Senegal orally exposed to 105 PFU/ml for Brazilian ZIKV (ZIKVT3FBra) and 104 PFU/ml for Senegalese strains (ZIKV132912) frozen for a week at – 80 °C.”.
- Line 267: "such as ... different glycosylation. Any evidence or proof?
Response: We listed this aspect as limitation because using different cells line may induce different glycosylation profile but we didn’t measure it during this study. Thus, we choose to remove this aspect and limit the sentence “…such as virus stocks produced on different cell lines,”. see line 293.
- Line 268: "virus stock lower than at least some human viremia level" Any evidence or proof of the comparison?
Response: Regarding the reviewer’s remark we choose to remove this comparison beyond the limitations of the study because even if we can find, in few cases, a human viremia level higher than those we used to infect mosquitoes, the mean of peak of ZIKV viremia in human is in the same range than our lowest titer of 104 PFU/ml [1], see line 293.
- Line 269: "virus titers different..." How is this information crucial?
Response: We considered here, that using different virus titers for different experiments can make comparison more difficult between ZIKV strains. We rephrase the sentence as following in line 293 “using different virus titers for different ZIKV strains”.
- Lines 284-285: "suspicion..." Suggest to add more discussion on the immune mechanism of Ae. mosquito that is involved in controlling the virus infection.
Response: Following the reviewer remarks, we have expended the discussion to add the following sentences for more arguments on the immune mechanism “Mosquitoes, as the vectors of several diseases, are susceptible to pathogen infection during their life cycles and use innate immune system to fight against. The innate immune system, including the production of antimicrobial peptides and lysozymes, phagocytosis, and melanization plays a significant role in limiting viruses to a non-lethal level [2]. A more recent study showed that ZIKV induced RNA interference response in Ae. aegypti [3].”. See lines 309-314.
- Lines 287-291: Results were reiterated instead of critical discussion on the observation/data. Improve these lines. Authors may explain why the decline occurred alongside similar evidence from previous studies.
Response: According to the reviewer comment we rephrased the paragraph to show how the observation/data support the hypothesis of mosquito immune response, lines 314-317. Below, the updated paragraph: “This suspicion that the immune system controls the infection is strongly supported by the dissemination rate of 25% observed at 5 dpe followed by the complete absence of dissemination observed at 10 dpe (when the decline in infection rate was noted for ZIKV T3FBra strain).”
- Lines 306-310: Split the information into separate sentences so that they clearly present the ideas.
Response: Following to the reviewer request the information has been split into 2 separate sentences as following “The same titer of 104 FFU/ml for the Senegalese ZIKV strain has previously shown infection, dissemination and transmission rates of up to 80%, 70% and 40% for Ae. aegypti mosquitoes from Brazil and 40%, 30% and 10% for Ae. aegypti mosquitoes from Texas, USA. This viral titer showed also infection, dissemination and transmission rates of 100%, 100% and 20%, respectively, for Ae. aegypti mosquitoes from Dominican Republic.”. See lines 346-351.
- Lines 316-319: Any specific reasons why exotic Ae. mosquitoes are more competent?
Response: In the same experiment using the same Zika virus strains, exotic Ae. aegypti mosquitoes were more competent. The specific reasons could be mainly genetics but also other factors. We know that the vector competence may change significantly from one mosquito population to other belonging to the same species. Ae. aegypti formosus, the ancestral sylvatic and zoophilic form native to Africa, is the most common form in West Africa. We integrated the following sentence to add more discussion about geographical difference of vector competence “Likewise, several vector competence studies have shown that Ae. aegypti from West Africa are more refractory for arboviruses like dengue and yellow fever viruses compared to Ae. aegypti from America or Asia [4–6].”, see lines 360-362.
- Lines 333: Future studies in what perspectives? Compatibility of receptors of the mosquitoes with the virus?
Response: We remove this paragraph in line 375 to add at the end of discussion the following sentence to mention the need of more studies concerning both mosquito species “about genetic factors (existence or not of barriers for the virus) or mosquito-virus-microbiome interactions may be the cause of this incompetency of Senegalese vectors to transmit ZIKV. “, see lines 390-407.
- Grammatical errors: as attached, lines 337, 339, and 347.
Response: According to the reviewer suggestion the grammatical errors have been corrected. See lines 377, 379, 383, 386 and 387 as following “Our results showed that infection of both ZIKV strains in Ae. aegypti from Dakar (Table 4) was not vertically transmitted to the next generation (Table 5). These results differ from those obtained in a recent study in which a population of Ae. aegypti from Thailand was able to transmit ZIKV to the next generation but at a very low rate of 0.3% (1/290) [7]. Given this very low rate, the absence of vertical transmission in our study with the Senegalese Ae. aegypti population may be due to the small number of specimens tested (n=283). Moreover, Ae. aegypti populations from other biogeographic areas also showed very low rates of vertical transmission of flaviviruses, with rates ranging from 0.21% (1: 472) to 0.15% (1: 632) for yellow fever virus (VFJ) [8], from 0.24% (1/401) to less than 0.0005% (1/1700) for DENVs [9], and from 1.61% (1/62) to 1.38% (1/72) for WNV [10]”.
We thank the reviewer for the attention he gives to our manuscript. We have taken into account of his preoccupation and have addressed his different concerns in the revised version; his remarks have improved the quality of the paper.
References:
- Lequime, S.; Dehecq, J.-S.; Matheus, S.; de Laval, F.; Almeras, L.; Briolant, S.; Fontaine, A. Modeling Intra-Mosquito Dynamics of Zika Virus and Its Dose-Dependence Confirms the Low Epidemic Potential of Aedes Albopictus. PLoS Pathog. 2020, 16, e1009068, doi:10.1371/journal.ppat.1009068.
- Xi, Z.; Ramirez, J.L.; Dimopoulos, G. The Aedes Aegypti Toll Pathway Controls Dengue Virus Infection. PLoS Pathog. 2008, 4, e1000098, doi:10.1371/journal.ppat.1000098.
- Saldaña, M.A.; Etebari, K.; Hart, C.E.; Widen, S.G.; Wood, T.G.; Thangamani, S.; Asgari, S.; Hughes, G.L. Zika Virus Alters the MicroRNA Expression Profile and Elicits an RNAi Response in Aedes Aegypti Mosquitoes. PLoS Negl. Trop. Dis. 2017, 11, e0005760, doi:10.1371/journal.pntd.0005760.
- Miller, B.R.; Mitchell, C.J. Genetic Selection of a Flavivirus-Refractory Strain of the Yellow Fever Mosquito Aedes Aegypti. Am. J. Trop. Med. Hyg. 1991, 45, 399–407, doi:10.4269/ajtmh.1991.45.399.
- Tabachnick, W.J.; Wallis, G.P.; Aitken, T.H.G.; Miller, B.R.; Amato, G.D.; Lorenz, L.; Powell, J.R.; Beaty, B.J. Oral Infection of Aedes Aegypti with Yellow Fever Virus: Geographic Variation and Genetic Considerations. Am. J. Trop. Med. Hyg. 1985, 34, 1219–1224, doi:10.4269/ajtmh.1985.34.1219.
- Bosio, C.F.; Beaty, B.J.; Black, W.C. Quantitative Genetics of Vector Competence for Dengue-2 Virus in Aedes Aegypti. Am. J. Trop. Med. Hyg. 1998, 59, 965–970, doi:10.4269/ajtmh.1998.59.965.
- Thangamani, S.; Huang, J.; Hart, C.E.; Guzman, H.; Tesh, R.B. Vertical Transmission of Zika Virus in Aedes Aegypti Mosquitoes. Am. J. Trop. Med. Hyg. 2016, 95, 1169–1173, doi:10.4269/ajtmh.16-0448.
- Beaty, B.J.; Tesh, R.B.; Aitken, T.H. Transovarial Transmission of Yellow Fever Virus in Stegomyia Mosquitoes. Am. J. Trop. Med. Hyg. 1980, 29, 125–132, doi:10.4269/ajtmh.1980.29.125.
- Rosen, L.; Shroyer, D.A.; Tesh, R.B.; Freier, J.E.; Lien, J.C. Transovarial Transmission of Dengue Viruses by Mosquitoes: Aedes Albopictus and Aedes Aegypti. Am. J. Trop. Med. Hyg. 1983, 32, 1108–1119, doi:10.4269/ajtmh.1983.32.1108.
- Baqar, S.; Hayes, C.G.; Murphy, J.R.; Watts, D.M. Vertical Transmission of West Nile Virus by Culex and Aedes Species Mosquitoes. Am. J. Trop. Med. Hyg. 1993, 48, 757–762, doi:10.4269/ajtmh.1993.48.757.

Round 2
Reviewer 1 Report
Authors have significantly improved the manuscript and Recommend acceptance.